# International epidemiology databases to evaluate AIDS (IeDEA) in sub-Saharan Africa, 2012–2019

Frédérique Chammartin,[1] Cam Ha Dao Ostinelli,[1] Kathryn Anastos,[2] Antoine Jaquet,[3] Ellen Brazier,[4,5] Steven Brown,[6] Francois Dabis,[3] Mary-Ann Davies,[7] Stephany N Duda,[8] Karen Malateste,[3] Denis Nash,[4,5] Kara Wools-Kaloustian,[9] Per M von Groote,[1] Matthias Egger [iD] [1,7]

For numbered affiliations see end of article.

**Correspondence to**
Matthias Egger;
matthias.egger@ispm.unibe.ch

## ABSTRACT

**Purpose** The objectives of the International epidemiology databases to evaluate AIDS (IeDEA) are to (i) evaluate the delivery of combination antiretroviral therapy (ART) in children, adolescents and adults in sub-Saharan Africa, (ii) to describe ART regimen effectiveness, durability and tolerability, (iii) to examine HIV-related comorbidities and coinfections and (iv) to examine the pregnancy-related and HIV-related outcomes of women on ART and their infants exposed to HIV or ART in utero or via breast milk.

**Participants** IeDEA is organised in four regions (Central, East, Southern and West Africa), with 240 treatment and care sites, six data centres at African, European and US universities, and almost 1.4 million children, adolescents and adult people living with HIV (PLWHIV) enrolled.

**Findings to date** The data include socio-demographic characteristics, clinical outcomes, opportunistic events, treatment regimens, clinic visits and laboratory measurements. They have been used to analyse outcomes in PLWHIV-1 or PLWHIV-2 who initiate ART, including determinants of mortality, of switching to second-line and third-line ART, drug resistance, loss to follow-up and the immunological and virological response to different ART regimens. Programme-level estimates of mortality have been corrected for loss to follow-up. We examined the impact of coinfection with hepatitis B and C, and the epidemiology of different cancers and of (multidrug resistant) tuberculosis, renal disease and of mental illness. The adoption of 'Treat All', making ART available to all PLWHIV regardless of CD4+ cell count or clinical stage was another important research topic.

**Future plans** IeDEA has formulated several research priorities for the 'Treat All' era in sub-Saharan Africa. It recently obtained funding to set up sentinel sites where additional data are prospectively collected on cardiometabolic risks factors as well as mental health and liver diseases, and is planning to create a drug resistance database.

## Strengths and limitations of this study

► An important strength of the International epidemiology databases to evaluate AIDS (IeDEA) cohort collaboration in sub-Saharan Africa is its large size, which allows analyses of outcomes of antiretroviral therapy (ART) in children, adolescents and pregnant and postpartum women, across diverse settings.

► The data reflect routine care across a wide range of real-world settings during the scale up of ART in sub-Saharan Africa and thus provide a valuable platform to conduct operational and clinical research and to study temporal trends and the impact of changes in guidelines and other interventions.

► The development of a standardised Data Exchange Standard protocol has contributed to increase data quality, and data have been enriched by linkage to cancer registries, vital registries and administrative databases.

► Collaborations with the WHO, United Nations Programme on AIDS, the mathematical modelling community and other consortia have ensured that the analyses of the African IeDEA regions contributed to global health policy and decision-making.

► Weaknesses include the limitations inherent in secondary use of routine clinical care data, with missing data, the lack of standardised follow-up visits and substantial loss to follow-up resulting in unknown outcome.

## INTRODUCTION

The roll-out of combination antiretroviral therapy (ART) in sub-Saharan Africa from 2004 onwards has substantially improved the prognosis of HIV-1 infection, with a decline in AIDS-related deaths[1] and a decline in the incidence of new HIV-1 infections.[2] However, in many settings, HIV/AIDS is still a public health threat. An estimated 1.8 million new infections occurred in 2017 and almost a million adult and child deaths were due to HIV, most of them in sub-Saharan Africa.[2]

The WHO, the Joint United Nations Programme on HIV/AIDS (UNAIDS) and many of the countries most heavily affected by the HIV epidemic have committed to ending HIV/AIDS as a major public health problem by 2030.[1] Targets to be reached by 2020 include that 90% of people living with

HIV (PLWHIV) be aware of their status, 90% of those diagnosed initiate ART and 90% of those on ART achieve undetectable viral loads (the 90-90-90 targets).[3] Progress towards these goals has been more substantial in Eastern and Southern Africa than in West and Central Africa. Of the 20.6 million PLWHIV in Eastern and Southern Africa, an estimated 58% were virally suppressed, compared with 39% of 5.0 million in West and Central Africa.[4]

Established more than 10 years ago by the National Institutes of Health (NIH), the International epidemiology databases to evaluate AIDS (IeDEA) are a global cohort collaboration that collects HIV/AIDS data from HIV care and treatment programmes, including in sub-Saharan Africa. The regional IeDEA data centres consolidate, curate and analyse data to evaluate the outcomes of PLWHIV/PLWAIDS and monitor progress. The first years of the cohorts in sub-Saharan Africa were described previously;[5] here we provide an update on methods, key data and future plans.

## COHORT DESCRIPTION

In 2006, the National Institute of Allergy and Infectious Diseases sought applications for a global consortium structured through regional centres to pool clinical and epidemiological data on PLWHIV, in order to address questions that could not be answered by individual cohorts.[5] IeDEA covers seven geographic regions, namely North America, the Caribbean and Central/South America, the Asia-Pacific and four regions in sub-Saharan Africa: West Africa, Central Africa, East Africa and Southern Africa. The project was initially funded for a 5-year period and has since been extended twice, with the current funding cycle ending in 2021.

### Settings and number of PLWHIV enrolled

To date, the African regions of IeDEA received data from 240 HIV care and treatment facilities in 19 sub-Saharan African countries (figure 1A). Close to 1 400 000 PLWHIV who initiated ART in sub-Saharan Africa are included (figure 1B), of whom over 680 000 are currently in care. In East and Southern Africa, both urban and rural facilities are well represented, while in Central and West Africa, urban facilities dominate. Facilities are predominately public (94%) and operated at the primary or secondary care level, with the exception of West Africa where 70% of facilities are at the tertiary level of care (table 1).

### Data collection at individual and site level

Since its inception, IeDEA has collected routine clinical data of PLWHIV followed under treatment, which includes sociodemographic characteristics, clinical outcomes, opportunistic events, treatment regimens, clinic visits and laboratory measurements. More recently, the IeDEA network has developed a Data Exchange Standard protocol (see www.iedeades.org) with 25 data tables, which include a total of 228 unique variables (36 compulsory and 192 additional variables) and within-region unique patient research identifiers. Standardised data collection is supported by eight codebooks, including the Anatomical Therapeutic Chemical (ATC) classification for drugs and lists with codes for reasons for stopping treatment, for dropping out of the cohort, for mode of HIV infection, country, type and site of comorbidities, laboratory measurements and units of measurements and type of viral load assay. The collection, management and sharing of data is facilitated by the Harmonist toolkit, a software and standards package that supports research

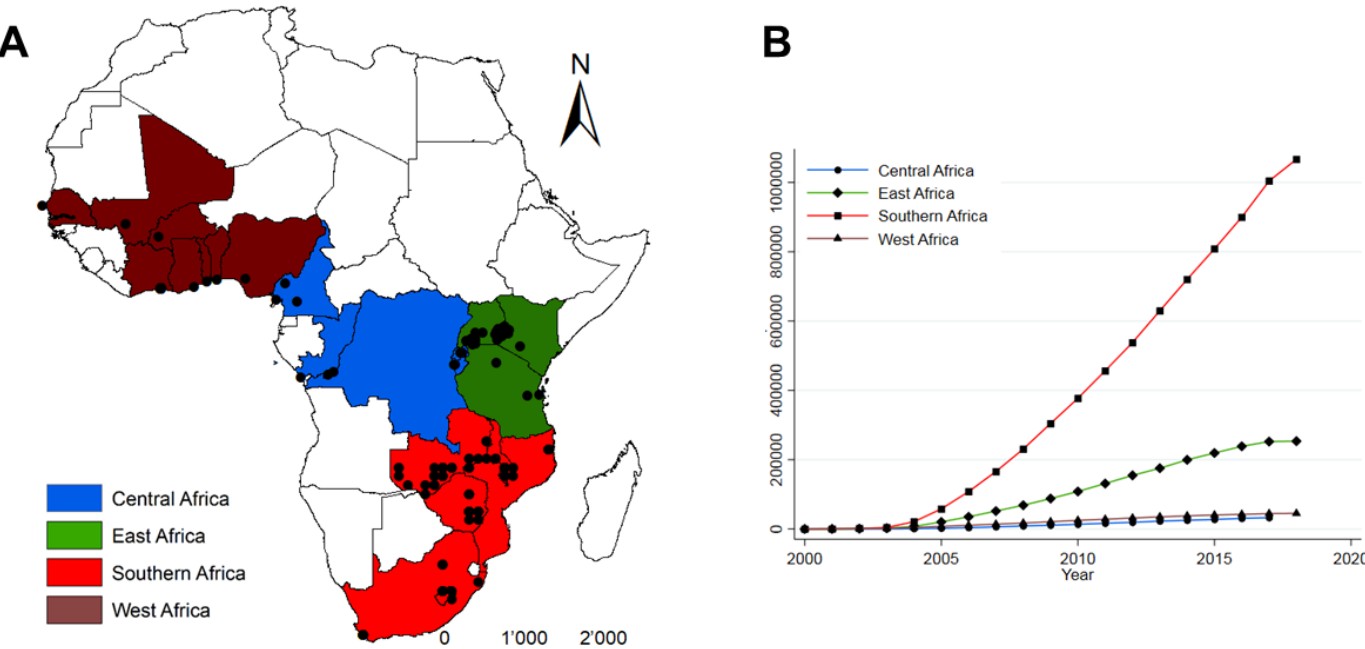

**Figure 1** Map of the 240 active facilities participating in the four African regions of the International epidemiology Databases to Evaluate AIDS (A), together with cumulative numbers of patients starting antiretroviral therapy (B).

**Table 1** Characteristics of 240 facilities providing ART in the African regions of the IeDEA (source: site assessment survey 2017 and IeDEA database 2019)

| | West Africa | Central Africa | East Africa | Southern Africa | All regions (%) |
|---|---|---|---|---|---|
| **No. of active facilities** | 17 | 19 | 72 | 132 | 240 |
| **No. of patients on ART** | 45 015 | 32 754 | 252 266 | 1 066 591 | 1 396 626 |
| **Location** | | | | | |
| Urban | 17 | 19 | 19 | 65 | 120 (50) |
| Rural | 0 | 0 | 52 | 67 | 119 (50) |
| Missing | 0 | 0 | 1 | 0 | 1 (0) |
| **Level of care** | | | | | |
| Primary | 4 | 12 | 49 | 107 | 172 (72) |
| Secondary | 1 | 0 | 16 | 20 | 37 (15) |
| Tertiary | 12 | 7 | 6 | 5 | 30 (13) |
| Missing | 0 | 0 | 1 | 0 | 1 (0) |
| **Type of facility** | | | | | |
| Public | 12 | 16 | 69 | 128 | 225 (94) |
| Private | 2 | 3 | 3 | 4 | 12 (5) |
| Missing | 3 | 0 | 0 | 0 | 3 (1) |
| **Viral load** | | | | | |
| Routine testing | 12 | 16 | 46 | 74 | 148 (62) |
| Tests performed onsite | 8 | 6 | 4 | 3 | 21 (9) |
| Tests performed offsite | 6 | 13 | 53 | 127 | 199 (83) |
| **CD4 monitoring** | | | | | |
| Routine testing | 14 | 6 | 9 | 97 | 126 (53) |
| Tests performed onsite | 11 | 8 | 23 | 51 | 93 (39) |
| Tests performed offsite | 3 | 11 | 35 | 71 | 120 (50) |
| **HIV-1 genotypic drug resistance** | | | | | |
| Routine testing | 2 | 2 | 10 | 48 | 62 (26) |
| **Routine tracing of patients lost to follow-up** | | | | | |
| Yes | 13 | 19 | 56 | 93 | 181 (75) |
| No | 1 | 0 | 1 | 37 | 39 (16) |
| Missing | 3 | 0 | 15 | 2 | 20 (8) |
| **Tracing method*** | | | | | |
| Phone | 14 | 18 | 57 | 110 | 199 (83) |
| Text message/mail/email | 2 | 2 | 10 | 14 | 28 (15) |
| Home visit | 9 | 17 | 52 | 129 | 207 (86) |
| **Medication disruption/stock outs over last 12 months** | | | | | |
| First-line ART | 5 | 2 | 20 | 32 | 59 (25) |
| Second-line ART | 5 | 6 | 18 | 16 | 45 (19) |

*Sites may use more than one method.
ART, antiretroviral therapy; IeDEA, International epidemiology Databases to Evaluate AIDS.

projects through the Research Electronic Data Capture (REDCap) system.[6]

In recent years, site assessments and site surveys have been conducted on a regular basis to collect up-to-date information related to available clinical service and care models in the participating facilities. For example, a study compared the characteristics and comprehensiveness of adult HIV care and treatment programmes in sub-Saharan Africa with programmes in the Americas and Asia-Pacific region.[7] Other studies examined the

management of mental health and substance use disorders,[8] or the diagnostic and screening practices for (drug resistant) tuberculosis in adult and paediatric patients.[9–11] Furthermore, the routine data collected by participating sites have been enriched in some countries by linking the HIV databases to cancer registries,[12 13] vital registries[14] or administrative databases.[15]

### Trends in CD4 cell count and viral load measurements

While WHO continues to recommend a CD4 cell count before starting ART to inform the management of advanced disease and differentiated care in the Treat All era, it also recommends that CD4 testing be replaced by viral load measurement for monitoring of treatment and identification of treatment failure.[16] Figure 2 shows that in Southern and West Africa, the number of CD4 measurements tended to be stable over time, despite an increasing number of PLWHIV in care, while in East and Central Africa, the number of CD4 measurements dropped. At present, 53% of the active facilities reported routine CD4 testing and 62% routine viral load testing (table 1). The US President's Emergency Plan for AIDS Relief, which provides substantial funding for AIDS treatment, care and prevention in countries most affected by the epidemic, has progressively reduced its support for CD4 testing.[17]

### Trends in ART

Until recently, the recommended first-line ART regimen in sub-Saharan Africa consisted of two nucleoside reverse transcriptase inhibitors and one non-nucleoside reverse transcriptase inhibitor (2NRTIs+1NNRTI). The combination of tenofovir (TDF), lamivudine (3TC) (or emtricitabine (FTC)) and efavirenz (EFV) is the current treatment of choice. The phasing out of stavudine (D4T) and nevirapine (NVP) was almost complete in 2014 (table 2). East Africa and Southern Africa are currently rolling out dolutegravir, an integrase inhibitor with a high barrier to resistance.[18 19] Due to concerns about an increased risk of neural tube defects if taken during pregnancy,[20] the roll-out to women has been delayed or limited in some settings. Of note, drug stock-outs in the last 12 months were reported by 59 facilities for first-line drugs, and by 45 for second-line drugs (table 1).

### Mortality and retention in care

In cohorts of PLWHIV who initiated ART in consecutive 2-year periods from 2001 to 2016, mortality at 3 years declined substantially in all African IeDEA regions (figure 3A). Loss to follow-up, defined as more than 90 days late to the next scheduled visit, remained substantial in all regions, and particularly high in Southern Africa (figure 3B). Retention in care is key to the success of the public health approach to ART. Loss to follow-up has been an important issue for IeDEA, and activities to trace PLWHIV not returning to the clinic have increased in recent years. At present, tracing of PLWHIV on ART who were lost to follow-up is in place in 89% of the active facilities; 75% have implemented it routinely (table 1). Tracing methods vary widely across facilities and include phone calls and home visits by clinic staff or community health workers.

### Patient and public involvement

IeDEA is based on the collection of routine clinical data and no patients were involved in developing the research question, outcome measures and overall design of the collaboration. Due to the anonymous nature of the data, we cannot disseminate the results of analyses of the data directly to study participants.

## FINDINGS TO DATE

Over 500 publications in MEDLINE acknowledge funding from a core grant from the NIH to one or several African IeDEA regions, and these publications have been cited over 10 000 times. Multiregional projects are developed in IeDEA working groups, which currently address eight clinical areas: (i) cancer, (ii) ART outcomes, (iii) hepatitis, (iv) mental health, (v) mother–infant and paediatrics, (vi) renal disease, (vii) substance use and (viii) tuberculosis. Multiregional research concepts are discussed in the Executive Committee of IeDEA, revised and approved or rejected. In recent years, several analyses were done in collaboration with WHO or UNAIDS.[21–23] The number of publications reporting multiregional analyses from several African IeDEA regions increased over time, from one such publication in 2007 to 24 multiregional publications in 2018. Some of the key studies are summarised below, with a focus on more recent and on multiregional analyses.

### Treatment outcomes in adults, adolescents, children and pregnant women

Several studies examined outcomes in PLWHIV-1 or PLWHIV-2 who initiate ART, including determinants of mortality, of switching to second-line and third-line ART, drug resistance, loss to follow-up and the immunological and virological response to different ART regimens.[24–37] For example, the African IeDEA regions contributed importantly to a large-scale analysis of outcomes in adolescents living with perinatally acquired HIV, which showed that HIV-associated mortality during adolescence was substantially higher in sub-Saharan Africa, South and Southeast Asia, and South America and the Caribbean than in Europe.[36] A similar analysis of adolescents living with HIV showed that mortality and loss to follow-up were worse among those entering care at 15 years or older.[37] The authors concluded that adolescents must be evaluated separately from younger children and adults to identify population-specific reasons for death and loss to follow-up.[37]

### Programme-level mortality

It became clear early on during the scale-up of ART in sub-Saharan Africa, that loss to follow-up of patients initiating

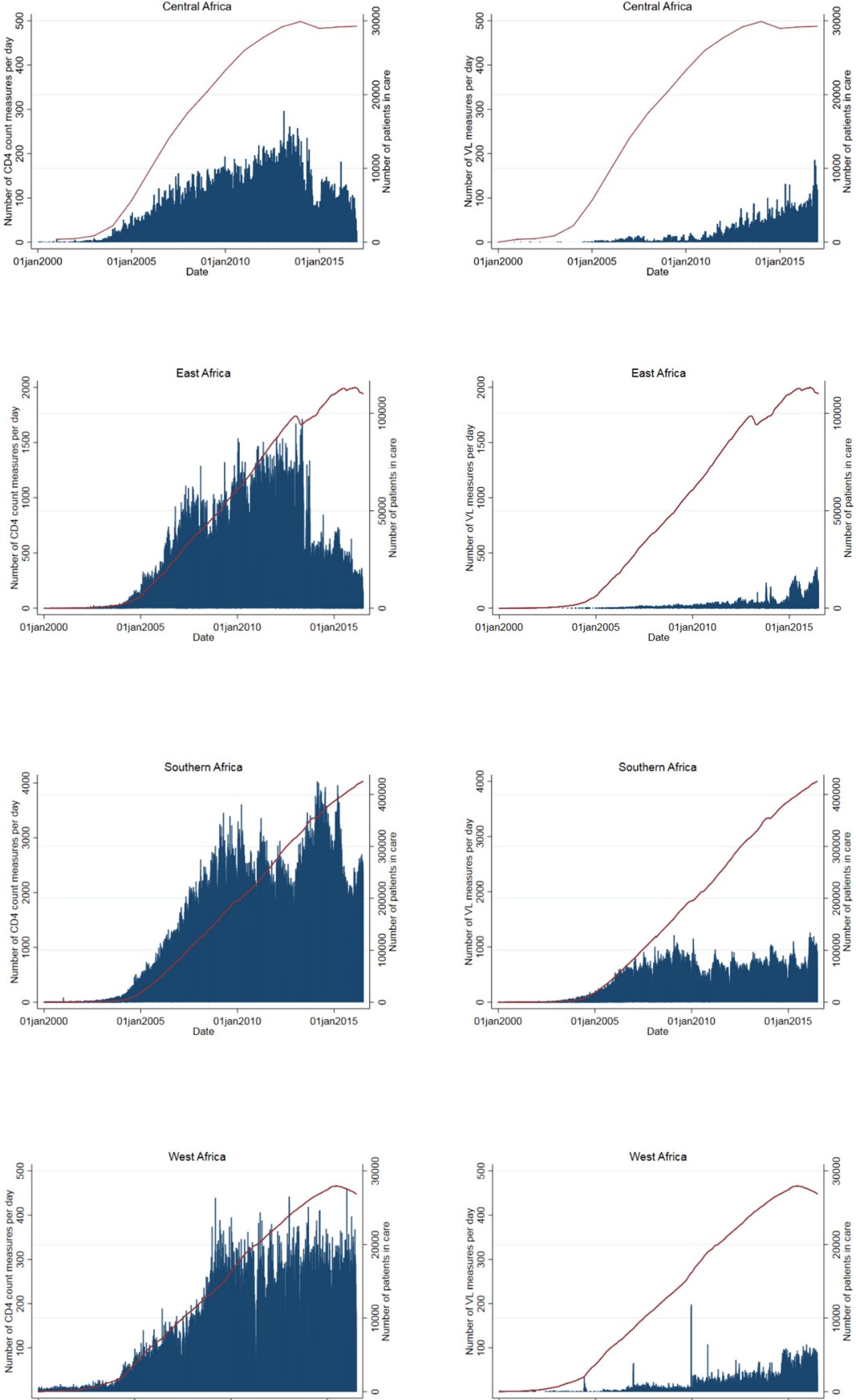

**Figure 2** Daily number of CD4 cell counts and viral load (VL) measurements over time (bar chart) and the number of patients in care (red line).

**Table 2** Proportion of patients with different nucleoside and non-nucleoside reverse transcriptase inhibitor regimens at the start of first-line antiretroviral therapy, by time period and region

### Central Africa

| | 2001-2002 | 2003-2004 | 2005-2006 | 2007-2008 | 2009-2010 | 2011-2012 | 2013-2014 | 2015-2016 |
|---|---|---|---|---|---|---|---|---|
| No of patients | – | 927 | 2766 | 4930 | 4848 | 5574 | 6048 | 5423 |
| **NRTI** | | | | | | | | |
| FTC+TDF | – | 1.2 | 4.4 | 0.8 | 3.5 | 5.0 | 6.6 | 9.2 |
| 3TC+TDF | – | 6.4 | 4.1 | 3.9 | 34.4 | 49.7 | 60.1 | 76.5 |
| 3TC+D4T | – | 60.0 | 55.6 | 33.7 | 8.8 | 2.7 | 0.1 | 0.0 |
| 3TC+AZT | – | 29.6 | 32.7 | 54.9 | 48.6 | 35.9 | 27.0 | 6.6 |
| 3TC+ABC | – | 1.1 | 1.4 | 2.1 | 2.8 | 4.1 | 5.1 | 7.3 |
| Other | – | 1.8 | 1.8 | 4.7 | 1.9 | 2.6 | 1.1 | 0.4 |
| **NNRTI** | | | | | | | | |
| NVP | – | 64.1 | 65 | 73.6 | 66.6 | 49.2 | 22.5 | 6.3 |
| EFV | – | 35.7 | 34.3 | 25.8 | 32.9 | 50.1 | 76.9 | 93.4 |
| Other | – | 0.2 | 0.7 | 0.7 | 0.5 | 0.7 | 0.6 | 0.2 |

### East Africa

| | 2001-2002 | 2003-2004 | 2005-2006 | 2007-2008 | 2009-2010 | 2011-2012 | 2013-2014 | 2015-2016 |
|---|---|---|---|---|---|---|---|---|
| No of patients | 819 | 6710 | 27 614 | 33 290 | 39 674 | 46 578 | 44 834 | 38 905 |
| **NRTI** | | | | | | | | |
| FTC+TDF | 0.8 | 5.4 | 0.3 | 1.0 | 0.9 | 1.1 | 1.4 | 0.7 |
| 3TC+TDF | 2.7 | 5.0 | 1.1 | 2.3 | 9.3 | 55.1 | 77.5 | 89.7 |
| 3TC+D4T | 78.9 | 80.0 | 83.2 | 54.1 | 37.4 | 3.0 | 0.5 | 0.1 |
| 3TC+AZT | 14.0 | 9.0 | 14.8 | 41.8 | 46.4 | 35.3 | 16.6 | 6.4 |
| 3TC+ABC | 0.6 | 0.3 | 0.4 | 0.8 | 5.9 | 5.4 | 4.0 | 3.2 |
| Other | 3.1 | 0.4 | 0.2 | 0.1 | 0.2 | 0.1 | 0.0 | 0.0 |
| **NNRTI** | | | | | | | | |
| NVP | 73.1 | 82.6 | 80.9 | 76.4 | 72.2 | 50.4 | 21.3 | 8.1 |
| EFV | 24.8 | 15.4 | 17.7 | 22.6 | 27.2 | 49.3 | 78.4 | 91.5 |
| Other | 2.0 | 2.0 | 1.4 | 1.0 | 0.6 | 0.3 | 0.4 | 0.4 |

### Southern Africa

| | 2001-2002 | 2003-2004 | 2005-2006 | 2007-2008 | 2009-2010 | 2011-2012 | 2013-2014 | 2015-2016 |
|---|---|---|---|---|---|---|---|---|
| No. of patients | 1345 | 19 434 | 86 775 | 122 953 | 146 425 | 161 014 | 182 330 | 179 424 |
| **NRTI** | | | | | | | | |
| FTC+TDF | 2.0 | 0.5 | 0.5 | 17.4 | 24.9 | 34.4 | 62.2 | 51.9 |
| 3TC+TDF | 2.5 | 1.1 | 1.1 | 2.1 | 17.1 | 33.5 | 22.2 | 40.3 |
| 3TC+D4T | 37.2 | 47.7 | 55.3 | 49.5 | 36.5 | 15.3 | 3.4 | 0.2 |
| 3TC+AZT | 47.0 | 39.7 | 27.7 | 16.2 | 10.7 | 8.7 | 6.8 | 2.5 |
| 3TC+ABC | 0.7 | 0.2 | 0.3 | 1.2 | 5.0 | 6.0 | 4.9 | 5.0 |
| Other | 10.6 | 10.9 | 15.1 | 13.6 | 5.8 | 2.2 | 0.5 | 0.1 |
| **NNRTI** | | | | | | | | |
| NVP | 51.1 | 58.1 | 64.8 | 55.4 | 47.2 | 36.7 | 15.6 | 3.7 |
| EFV | 43.7 | 37.8 | 31.4 | 41.5 | 50.0 | 61.2 | 82.4 | 93.9 |
| Other | 5.2 | 4.1 | 3.7 | 3.1 | 2.9 | 2.1 | 1.9 | 2.4 |

### West Africa

| | 2001-2002 | 2003-2004 | 2005-2006 | 2007-2008 | 2009-2010 | 2011-2012 | 2013-2014 | 2015-2016 |
|---|---|---|---|---|---|---|---|---|
| No. of patients | 822 | 3533 | 6348 | 6490 | 7748 | 6836 | 6136 | 4451 |
| **NRTI** | | | | | | | | |
| FTC+TDF | 0.6 | 0.0 | 0.1 | 1.5 | 11.5 | 17.6 | 11.5 | 11.6 |
| 3TC+TDF | 0.0 | 0.0 | 0.2 | 0.6 | 3.5 | 10.3 | 32.1 | 55.3 |
| 3TC+D4T | 39.2 | 46.8 | 51.6 | 42.6 | 21.6 | 4.8 | 0.6 | 0.0 |
| 3TC+AZT | 35.7 | 39.7 | 43.3 | 51.0 | 46.2 | 53.4 | 49.3 | 22.5 |
| 3TC+ABC | 0.2 | 0.1 | 0.4 | 0.9 | 0.9 | 4.1 | 5.8 | 10.3 |
| Other | 24.3 | 13.4 | 4.5 | 3.4 | 16.3 | 9.8 | 0.6 | 0.3 |
| **NNRTI** | | | | | | | | |
| NVP | 10.1 | 12.0 | 40.9 | 46.1 | 50.2 | 47.0 | 36.1 | 15.0 |
| EFV | 50.1 | 60.6 | 46.7 | 44.4 | 39.7 | 41.5 | 53.8 | 70.7 |
| Other | 39.8 | 27.4 | 12.5 | 9.6 | 10.2 | 11.5 | 10.1 | 14.3 |

ABC, abacavir; AZT, zidovudine; D4T, stavudine; EFV, efavirenz; FTC, emtricitabine; NNRTI, non-nucleoside reverse transcriptase inhibitor; NRTI, nucleoside reverse transcriptase inhibitor; NVP, nevirapine; 3TC, lamivudine; TDF, tenofovir.

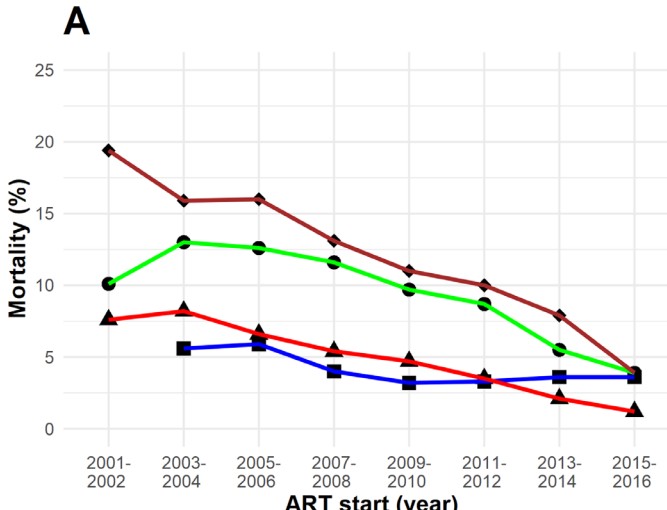

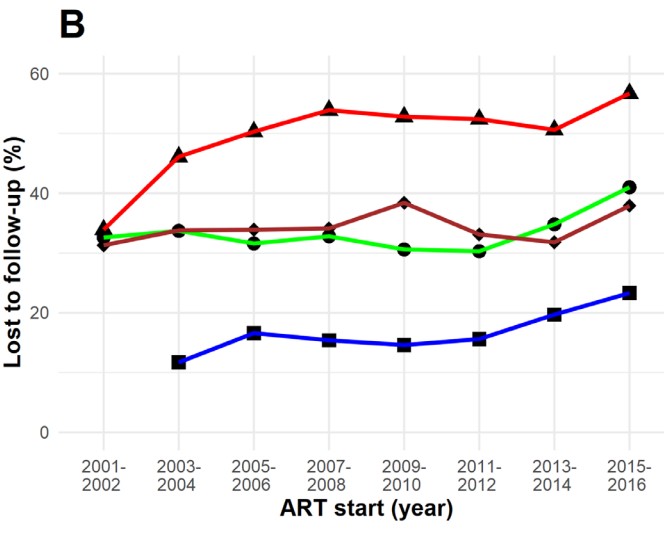

**Figure 3** Trends in mortality (A) and loss to follow (B), 2001 to 2016. ART, antiretroviral therapy.

ART was substantial,[38] and that mortality among patients lost was higher than among patients remaining in care.[39] Ignoring loss to follow-up might thus bias programme-level estimates of mortality, and much effort has gone into correcting programme-level mortality for loss to follow-up.[40–46] For example, an analysis of all four African regions showed that when analysing the uncorrected data observed in the clinics, 52% of adults and children were retained on ART, 42% were lost to follow-up and 6% had died 5 years after ART initiation.[46] After accounting for undocumented deaths and self-transfers, an estimated 67% of patients were retained on ART, 19% had stopped ART and 15% had died.[46]

### Coinfections and comorbidities

IeDEA investigators have examined the prevalence and impact of coinfection with hepatitis B and C, and the epidemiology of different cancers and of (multidrug resistant) tuberculosis, renal disease and of mental

illness.[8 47–58] For example, an analysis of IeDEA data and data from the Collaboration of Observational HIV Epidemiological Research in Europe (COHERE)[59] showed that children living with HIV from sub-Saharan Africa, but not those from Europe or Asia had a high risk of developing Kaposi sarcoma after starting ART.[58] Similarly, a recent analysis of IeDEA and COHERE data showed that compared with European women, rates of cervical cancer were 11 times higher in South Africa.[50] A recent multiregional study of multidrug resistant tuberculosis included HIV-positive and HIV-negative adults with tuberculosis from seven high-burden countries (Côte d'Ivoire, Democratic Republic of the Congo, Kenya, Nigeria, South Africa, Peru and Thailand). Molecular or phenotypic drug susceptibility testing was done locally and at a reference laboratory. The results showed that inaccurate local drug susceptibility testing led to undertreatment of drug-resistant tuberculosis and increased mortality.[54]

### Challenge of 'Treat All'

Nearly all countries in sub-Saharan Africa have now adopted national polices to offer ART to all PLWHIV regardless of CD4 cell count or clinical stage ('Treat All'), in order to meet the UNAIDS 90-90-90 targets. In 2011, Malawi was one of the first countries to implement such a strategy for the prevention of mother-to-child transmission, recommending ART for pregnant and breastfeeding women living with HIV, regardless of CD4 cell count or WHO clinical stage ('Option B+'). An IeDEA analysis of the Malawian experience showed that poor retention in care was a problem in many facilities, with early loss to follow-up particularly high in facilities with a high patient volume and in patients who start ART during pregnancy on the day of HIV diagnosis.[35] More recently, IeDEA investigators used regression discontinuity analysis to examine changes in rapid HIV treatment initiation after national 'Treat All' policy adoption in six countries (Burundi, Kenya, Malawi, Rwanda, Uganda and Zambia).[60] They showed a strong and sustained effect of the adoption of 'Treat All' policies on ART initiation within 30 days of enrollment in HIV care in all six countries.[60]

### Future plans

At the end of 2018, IeDEA published a consensus statement[61] and a journal supplement[62] on research priorities to inform the implementation of the 'Treat All' policy in children and adolescents,[63] pregnant and postpartum women,[64] and for mental health, substance use[65] and drug resistance.[66] These documents will guide IeDEA's future research agenda in sub-Saharan Africa. Furthermore, the creation of an IeDEA Sentinel Research Network will facilitate the collection of detailed data in selected IeDEA sites on cardiometabolic risk factors (eg, hypertension, diabetes and dyslipidaemia) liver disease (liver fibrosis and steatosis), mental health and substance use. In the East and Southern African regions, pharmacovigilance in pregnancy is being developed to assess the impact of ART on birth outcomes. A project involving all four African

regions studies the cascade of screening for cervical cancer, while the establishment of the South African HIV Cancer Match (SAM) study of over 10 million PLWHIV, from linkage of national laboratory with cancer registry data, will allow the study of less common cancers. Finally, the creation of a drug resistance database as a central repository for resistance tests performed in routine clinical care is another planned addition.

## COLLABORATIONS

The African regions of IeDEA have collaborated and continue to encourage collaborations with other consortia, cohort collaborations, the HIV modelling community and public health agencies as well as individuals wishing to use IeDEA data. Examples include work with COHERE,[50 58 67 68] the Collaborative Initiative for Paediatric HIV Education and Research[36 69] or the Measurement and Surveillance of HIV Epidemics consortium[22] as well as UNAIDS[22 23] and WHO.[21] Further collaborations are welcome. Investigators wishing to work with the IeDEA data should contact the teams at the regional data centres (see www.iedea.org for contact details) and send a concept sheet for the analyses they are interested in performing and the variables that would be required. Anyone wishing to work with IeDEA must sign a data-use agreement.

**Author affiliations**
[1]Institute of Social and Preventive Medicine, University of Bern, Bern, Switzerland
[2]Departments of Medicine and Epidemiology & Population Health, Albert Einstein College of Medicine, Bronx, New York, USA
[3]French National Research Institute for Sustainable Development (IRD), Inserm, UMR 1219, University of Bordeaux, Bordeaux, France
[4]Institute for Implementation Science in Population Health, City University of New York, New York, New York, USA
[5]Graduate School of Public Health and Health Policy, City University of New York, New York, New York, USA
[6]Department of Biostatistics, Indiana University School of Medicine, Indianapolis, Indiana, USA
[7]Centre for Infectious Disease Epidemiology and Research, School of Public Health and Family Medicine, University of Cape Town, Rondebosch, Western Cape, South Africa
[8]Department of Biomedical Informatics, Vanderbilt University School of Medicine, Nashville, Tennessee, USA
[9]Department of Medicine, Indiana University School of Medicine, Indianapolis, Indiana, USA

**Acknowledgements** The authors wish to thank the clinical and administrative staff at the participating clinics. We are grateful to all PLWHIV who contributed to the IeDEA database.

**Contributors** FC, PMvG and ME conceptualised the study. FC and CHDO performed statistical analyses. FC and ME wrote the first draft of the paper. CHDO, KA, AJ, EB, SB, FD, MAD, SND, KM, BSM, DN, KWK, PMvG and ME contributed to interpreting the data and to the writing and revising of the manuscript.

**Funding** The International Epidemiology Databases to Evaluate AIDS (IeDEA) is supported by the US National Institutes of Health's National Institute of Allergy and Infectious Diseases, the Eunice Kennedy Shriver National Institute of Child Health and Human Development, the National Cancer Institute, the National Institute of Mental Health, the National Institute on Drug Abuse and Alcoholism, the National Institute of Diabetes and Digestive and Kidney Diseases, the Fogarty International Center, the National Library of Medicine and the Office of the Director: Central

Africa, U01AI096299; East Africa, U01AI069911; Southern Africa, U01AI069924; West Africa, U01AI069919. Informatics resources are supported by the Harmonist project, R24AI124872. ME was supported by special project funding (Grant No. 174281) from the Swiss National Science Foundation.

**Map disclaimer** The depiction of boundaries on this map does not imply the expression of any opinion whatsoever on the part of BMJ (or any member of its group) concerning the legal status of any country, territory, jurisdiction or area or of its authorities. This map is provided without any warranty of any kind, either express or implied.

**Competing interests** None declared.

**Patient and public involvement** Patients and/or the public were not involved in the design, or conduct, or reporting, or dissemination plans of this research.

**Patient consent for publication** Not required.

**Ethics approval** The Ethics Committee of the Canton of Bern, the Ethics Committee of the University of Cape Town and the local ethics committees or institutional review boards all approved the use of routine clinical data for research within the IeDEA collaboration. For studies requiring additional data collection, separate ethics approval and study-specific informed consent is sought.

**Provenance and peer review** Not commissioned; externally peer reviewed.

**Data availability statement** Data are available upon reasonable request. Investigators wishing to work with IeDEA data should contact the regional data centres (see www.iedea.org) and send a concept sheet for the analyses they are interested in performing and the variables that would be required. See www.iedeades.org for list of variables. Those wishing to work with the data must sign a data-use agreement.

**ORCID iD**
Matthias Egger http://orcid.org/0000-0001-7462-5132

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
