## [Reviewer comments · BMJ Open]

ARTICLE DETAILS

TITLE (PROVISIONAL)	Cohort profile: The International epidemiology Databases to Evaluate AIDS (IeDEA) in sub-Saharan Africa, 2012-2019
AUTHORS	Chammartin, Frédérique; Dao Ostinelli, Cam Ha; Anastos, Kathryn; Jaquet, Antoine; Brazier, Ellen; Brown, Steven; DABIS, FRANCOIS; Davies, Mary-Ann; Duda, Stephany N; Malateste, Karen; Nash, Denis; Wools-Kaloustian, Kara; von Groote, Per M; Egger, Matthias

VERSION 1 – REVIEW

REVIEWER	Peter Wesley Young U.S. Centers for Disease Control and Prevention, Kenya
REVIEW RETURNED	16-Dec-2019

GENERAL COMMENTS	This profile describes the sub-Saharan Africa component of the IeDEA network of clinical cohorts of HIV-infected persons, an NIH-funded consortium designed to evaluate ART, describe ART regimen effectiveness, durability and tolerability, examine HIV-related comorbidities and co-infections, and examine outcomes in pregnant women and their HIV-exposed and/or ARV-exposed infants. The profile both describes the basic structure of the network as well as some of the subject areas where the network has contributed meaningfully to our understanding of the epidemiology of HIV, to trends in the programmatic response and outcomes of treatment, as well as effectiveness of treatment and PMTCT. The sub-Saharan portion of the network was last profiled in a 2012 manuscript in the International Journal of Epidemiology. This profile describes some of the programmatic changes that have affected the cohorts, such as rapid ART initiation, reduced use of CD4 counts and increasing coverage of systems to trace lost-to-follow-up clients. The profile also discusses the ongoing efforts to better characterize true LTFU and mortality. Finally, the profile discusses future efforts to better describe chronic disease risk in cohorts exposed to ART over time. Specific issues: There is an assertion that Southern and Eastern Africa appear to be on track to achieve the 90-90-90 targets (page 4, lines 26-27). Based on the latest estimates (e.g. UNAIDS Data 2019 report Fig 10.8 page 25) it would seem better to say that although tremendous progress has been made, it seems unlikely that the region will reach the 90-90-90 targets by 2020, with a gap of 1.1 million to reach the 1st 90 and 3.0 million to reach the 2nd and 3rd 90, respectively. Alternatively, if the authors are aware of a recent
--

	study or projection showing that the region is indeed on track to meet the targets, that could be cited. There is no statement on informed consent. If informed consent is not sought, or only sought for specific projects beyond routine data collection, that should ideally be described in the section on patient involvement (page 6, lines 30-35). Table 1 appears to have some inconsistencies, for example the number of public (225) + private (12) facilities does not equal the total (240), and only 181 + 39 = 220 have a response to whether LTFU tracing is performed. Is this due to missing data? Including a row for each indicator saying how many were missing, or at minimum a footnote indicating that the totals do not match the sum of the sub-totals due to missing values, would be helpful.
--	---

REVIEWER	Abigail Kroch Ontario HIV Treatment Network, Canada
REVIEW RETURNED	20-Dec-2019

GENERAL COMMENTS	No additional comments. Manuscript is clearly written and a good description of the cohort study.
---

REVIEWER	Patricia Rojas Sanchez Institute of Applied Health Research University of Birmingham
REVIEW RETURNED	13-Jan-2020

GENERAL COMMENTS	The leDEA in Africa provide clinical and epidemiological information for children, adolescents and adult people living with HIV. This platform is highly relevant to evaluate the effectiveness of the ART in sub-Saharan Africa. The authors did a lot of work. However, the resolution and quality of the data should be revised. As in this paper the authors described an extension and update of the cohort previously publish, the methods and the main outcomes are difficult to understand. In my opinion, the manuscript is not ready for publication yet, and it requires a major revision. Comment 1: According with the BMJ Open instructions for Cohort Profiles, the following items should be included: Introduction, Cohort description, Findings to date, Strengths and limitations, Collaboration, Further details. However, in the paper the authors have not included: Findings to date, Strengths and limitations. Could you please follow the instructions? Comment 2: It is not clear enough the total number of participants. Could be possible to add the number of patients in each country or facility? Comment 3: The main findings of this study are included in the area "KEY RESEARCH AREAS AND PUBLICATIONS ", but they are not very clear. As suggested previously, authors should include a section entitled "Findings to date" to explain the most notable results of the cohort and to define clearly the main outcomes of the study.
---

	Comment 4: It is not clear if research ethics (e.g. participant consent, ethics approval) are addressed appropriately
--	---

VERSION 1 – AUTHOR RESPONSE

Reviewer(s)' Comments to Author:

Reviewer: 1

Reviewer Name: Peter Wesley Young

Institution and Country:

U.S. Centers for Disease Control and Prevention, Kenya Please state any competing interests or state 'None declared': None declared

This profile describes the sub-Saharan Africa component of the leDEA network of clinical cohorts of HIV-infected persons, an NIH-funded consortium designed to evaluate ART, describe ART regimen effectiveness, durability and tolerability, examine HIV-related comorbidities and co-infections, and examine outcomes in pregnant women and their HIV-exposed and/or ARV-exposed infants. The profile both describes the basic structure of the network as well as some of the subject areas where the network has contributed meaningfully to our understanding of the epidemiology of HIV, to trends in the programmatic response and outcomes of treatment, as well as effectiveness of treatment and PMTCT.

The sub-Saharan portion of the network was last profiled in a 2012 manuscript in the International Journal of Epidemiology. This profile describes some of the programmatic changes that have affected the cohorts, such as rapid ART initiation, reduced use of CD4 counts and increasing coverage of systems to trace lost-to-follow-up clients. The profile also discusses the ongoing efforts to better characterize true LTFU and mortality. Finally, the profile discusses future efforts to better describe chronic disease risk in cohorts exposed to ART over time.

Thank you for this summary.

Specific issues:

There is an assertion that Southern and Eastern Africa appear to be on track to achieve the 90-90-90 targets (page 4, lines 26-27). Based on the latest estimates (e.g. UNAIDS Data 2019 report Fig 10.8 page 25) it would seem better to say that although tremendous progress has been made, it seems unlikely that the region will reach the 90-90-90 targets by 2020, with a gap of 1.1 million to reach the 1st 90 and 3.0 million to reach the 2nd and 3rd 90, respectively. Alternatively, if the authors are aware of a recent study or projection showing that the region is indeed on track to meet the targets, that could be cited.

Thank you. We have rephrased this section as follows and now cite the UNAIDS Data 2019 report:

Progress towards these goals has been more substantial in Eastern and Southern Africa than in West and Central Africa. Of the 20.6 million PLWHIV in Eastern and Southern Africa, an estimated 58% were virally suppressed, compared to 39% of 5.0 million PLWIV in West and Central Africa [4].

There is no statement on informed consent. If informed consent is not sought, or only sought for specific projects beyond routine data collection, that should ideally be described in the section on patient involvement (page 6, lines 30-35).

This statement is now included in the 'Further details' section. See also our response to the editor above.

Table 1 appears to have some inconsistencies, for example the number of public (225) + private (12) facilities does not equal the total (240), and only 181 + 39 = 220 have a response to whether LTFU tracing is performed. Is this due to missing data? Including a row for each indicator saying how many were missing, or at minimum a footnote indicating that the totals do not match the sum of the sub-totals due to missing values, would be helpful.

Thank you. Indeed, inconsistencies are due to missing data. We have now revised Table 1 to incorporate missing information.

Reviewer: 2

Reviewer Name: Abigail Kroch

Institution and Country: Ontario HIV Treatment Network, Canada Please state any competing interests or state 'None declared': None declared

No additional comments. Manuscript is clearly written and a good description of the cohort study.

Thank you.

Reviewer: 3

Reviewer Name: Patricia Rojas Sanchez

Institution and Country:
Institute of Applied Health Research
University of Birmingham

Please state any competing interests or state 'None declared': None declared

The leDEA in Africa provide clinical and epidemiological information for children, adolescents and adult people living with HIV. This platform is highly relevant to evaluate the effectiveness of the ART in sub-Saharan Africa.

The authors did a lot of work. However, the resolution and quality of the data should be revised. As in this paper the authors described an extension and update of the cohort previously published, the methods and the main outcomes are difficult to understand. In my opinion, the manuscript is not ready for publication yet, and it requires a major revision.

Thank you. We are not sure what the referee means with 'revising the data', but please note that we discuss the limitations of routine clinical data and their quality under 'Strengths and limitations':

"Weaknesses include the limitations inherent in secondary use of routine clinical care data, with missing data, the lack of standardised follow-up visits, and substantial loss to follow-up resulting in unknown outcome."

Comment 1: According with the BMJ Open instructions for Cohort Profiles, the following items should be included: Introduction, Cohort description, Findings to date, Strengths and limitations, Collaboration, Further details. However, in the paper the authors have not included: Findings to date, Strengths and limitations. Could you please follow the instructions?

Thank you. We included the sections 'Introduction', 'Cohort description' and 'Collaboration'. We erroneously called the section on findings to date 'Key research areas and publications'. We have changed this to 'Findings to date'. Also, we called the section on further details 'Footnotes'. Again, we now changed this to 'Further details'. The strengths and limitations were in the Box.

Comment 2: It is not clear enough the total number of participants. Could be possible to add the number of patients in each country or facility?

Thank you for this comment. We have added the number of patients to Tables 1 and 2.

Comment 3: The main findings of this study are included in the area "KEY RESEARCH AREAS AND PUBLICATIONS ", but they are not very clear. As suggested previously, authors should include a section entitled "Findings to date" to explain the most notable results of the cohort and to define clearly the main outcomes of the study.

We acknowledge that we used the wrong heading for this section. We have changed this. However, we feel the renamed section gives a good overview of the findings, with a focus on recent and multiregional analyses, structured by topic.

Comment 4: It is not clear if research ethics (e.g. participant consent, ethics approval) are addressed appropriately

This statement is now included in the Further details section. See also our response to the editor above.

VERSION 2 – REVIEW

REVIEWER	Peter Wesley Young U.S. Centers for Disease Control and Prevention Kenya
REVIEW RETURNED	02-Mar-2020

GENERAL COMMENTS	Thank you for addressing all of my prior comments. I believe this manuscript is ready for publication.
--

REVIEWER	Patricia Rojas University of Birmingham, UK
REVIEW RETURNED	24-Feb-2020

GENERAL COMMENTS	No additional comments in this review. Most of the comments have been included or clarified in the response letter. In my opinion the manuscript is ready for publication.
--